# Association between maternal education and breast feeding practices in China: a population-based cross-sectional study

Kun Tang,[1] Hanyu Wang,[2] Shi Hui Tan,[3] Tong Xin,[4] Xueqi Qu,[5] Tianyu Tang,[1] Yuqi Wang,[2] Yuning Liu,[5] Junjian Gaoshan[6]

KT and HW are joint first authors.

¹Research Center for Public Health, Tsinghua University, Beijing, China
²Institute for Medical Humanities, Peking University Health Science Centre, Beijing, China
³School of Basic Medical Sciences, Peking University Health Science Centre, Beijing, China
⁴Department of Global Health, Peking University Health Science Centre, Beijing, China
⁵School of Public Health, Peking University Health Science Centre, Beijing, China
⁶United Nations Population Fund China Office, Beijing, China

**Correspondence to**
Dr Kun Tang;
tangk@tsinghua.edu.cn

## ABSTRACT

**Objective** To investigate the association between maternal education and breast feeding in the Chinese population, with a consideration of household income and health-seeking behaviours.

**Design** A nationally representative population-based cross-sectional study.

**Setting** 77 counties from 12 geographically distinct regions in China.

**Participants** 10 408 mothers with children from 0 to 12 months of age, aged 15–53 years old (mean: 29.15, SD: 5.11) were classified into primary school or below group (n=781), middle school group (n=3842), high school/vocational school group (n=1990), college or above group (n=3795), according to their highest completed education.

**Outcomes** Five breastfeeding outcomes, including early initiation of breast feeding (EIB), exclusive breast feeding (EBF) under 6 months, predominant breast feeding under 6 months, current breast feeding and children ever breast fed, were calculated based on the standardised questionnaire from the WHO and Wellstart International's toolkit for monitoring and evaluating breastfeeding activities.

**Results** The absolute risk of EIB and EBF in the lowest maternal education level was 64.85% and 26.53%, respectively, whereas the absolute risk of EIB and EBF in the highest maternal education level was 77.21% and 14.06%, respectively. A higher level of maternal education was positively associated with EIB (risk ratio (RR): 1.22; 95% CI: 1.12 to 1.30) and was inversely associated with EBF (RR: 0.59; 95% CI: 0.38 to 0.88). Stratified by household income, a positive association with EIB was observed only in the group with the highest household income and an inverse association with EBF was found in both low household and high household income groups.

**Conclusions** Mothers with a higher education were more likely to initiate early breast feeding when they were also from a high-income household while also being less likely to exclusively breast feed their babies. Routine and successful nursing is crucial for the health of infants and is influenced by maternal education. Future public health interventions to promote breast feeding should consider the issues related to the educational level of mothers.

## INTRODUCTION

Extensive research has reported the diverse and compelling advantages of breast feeding for

## Strengths and limitations of this study

► The present study is a nationally representative population-based study with a large sample size, and is based on the first survey conducted in China to specifically explore the social determinants of breast feeding.

► This is the first study to investigate the relationship between maternal education and comprehensive breastfeeding patterns on a national scale in China.

► We specified breastfeeding indicators and explored the associations separately and stratified by possible influential factors.

► Maternal education in this study represents the formal education gained through schooling and it may not reflect the actual health literacy of the mothers.

► Information was based on self-report in the survey and feeding practices were determined using the 'last-24-hour method', which tends to overestimate the prevalence of exclusive breast feeding as compared with the 'since birth method'.

infants, mothers, families and society.[1] Breastfeeding has become more common in the last few years because of numerous interventions conducted at multiple levels from the communities, hospitals, and society as a whole, but the breastfeeding rates remain below optimal levels recommended by the World Health Organization (WHO) in many countries.[2 3] According to a recent cross-sectional report of 55 counties from 30 provinces in China, the prevalence of mother ever breastfeed was 79.6%, and only 20.8% of 14,539 children surveyed were exclusively breastfed at 6 months.[4]

WHO recognises almost all mothers as biologically capable to breast feed, except in a small number of health conditions, and that the use of breast milk substitutes is justifiable.[5] Breastfeeding practices, however, are not merely biological issues but are also related to health behaviours and are influenced by multifactorial determinants, including historical, socioeconomic and cultural factors.[6] Women's breastfeeding practices are also affected by

personal attributes, such as age, weight, education and confidence.[6] In China, breastfeeding practices face unique challenges, due to China's diverse cultural values, political system and historical perception of breast feeding.[7 8]

The education status of the mother has been identified as an important determinant for child welfare, survival and health.[9–11] Skafida et al found that maternal education was a more informative predictor of breastfeeding initiation than occupation.[12] Although maternal education and breastfeeding practices have been studied extensively and most of the previous literature reported a positive association between maternal education and breastfeeding practices, the positive association may not be accurate in all contexts.[13–17] Studies conducted in Nepal, the United States and Italy showed that higher maternal education level was related to better breastfeeding practices.[13–15] However, studies in Ethiopia and Bangladesh observed a negative association between breast feeding and maternal education.[16 17]

Most of the previous studies conducted in China explored the effect of maternal education on breast feeding, together with other socioeconomic factors. The results across studies were different and sometimes reached opposing conclusions.[18] Using data from 12 regions in China, the aim of this study is to investigate the association between maternal education and breast feeding, taking into account household income and health-seeking behaviours. The findings of this study contribute to the growing body of evidence on the social determinants of breast feeding and have the potential to provide evidence for related interventions and policies.

## METHODS
### Study design and participants
This population-based study was conducted in 12 geographically defined regions in China from July 2017 to January 2018. Multi-stage sampling technique was adopted for the selection of the participants. In the first stage, population proportionate sampling method was applied, due to the population structure and the status of maternal and child health and socioeconomic development. The selection of the 12 study sites was made carefully, aiming to maximise geographical diversity (including northern, southern, eastern and western regions with very different climate and cultures), socioeconomic diversity (including affluent cities and impoverished inland rural areas, as well as cities of different population sizes) and taking into account the maternal and child health situation (ie, the maternal mortality rates), immigration population proportion, population stability, local commitment and capacity in order to provide a balanced and representative sampling. In the end, four megacities (Beijing in North China, Nanning of Guangxi Province in South Central China, Hefei of Anhui province and Nanjing of Jiangsu province in East China), four medium-sized cities (Longyan of Fujian province in East China, Liaoyuan of Jilin province in Northeast China, Honghe of Yunnan province and Luzhou of Sichuan province in Southwest China), two countryside areas

(Zhengding County of Hebei province in North China and Jia County of Henan province in Central China) and two underdeveloped rural areas (Tailai county of Heilongjiang province in Northeast China and Ledu county of Qinghai province in Northwest China) were selected. In the second stage, one district or county was chosen from the 12 regions by simple random sampling method. In the third stage, a simple sampling method was implemented to select four to eight communities or villages from the chosen district or county in the 12 regions. A survey team of five full-time staff with medical qualifications and field experience was established in each of the 12 regions. Mothers with children under 12 months were identified through birth registration by the China National Centre for Disease Control. Mothers whose children were under 12 months were eligible to participate in this study. To encourage involvement, incentives such as gift cards and simple household commodities were prepared for each participant. In the end, 10 408 mothers from 77 counties were recruited for the present study. After registration and obtaining informed consent, trained health staff with smartphone-based questionnaires collected participants' information. The questionnaire asked for mothers' sociodemographic information, socioeconomic information, breastfeeding behaviour and breastfeeding-related environment. Over 90% of the participants completed the entire questionnaire. The Institutional Review Board at Peking University Health Science Centre and the China National Centre for Disease Control approved this study. All participants included in this study provided written informed consent.

### Variable definitions
Maternal education was the primary variable in the study. In the baseline survey, maternal education was classified into eight categories, including no formal education, did not graduate from primary school, primary school, middle school, high/vocational school, associate/junior college, university, and postgraduate and above. In the present study, those with no formal school education, dropped out from primary school or those with only a primary school education were classified into the 'Primary School and Below' group; those with middle school education were classified as the 'Middle School' group; those with high/vocational school education were classified as the 'High/Vocational School' group; and those with an education of associate/junior college, university, postgraduate and above were classified as the 'College and Above' group. This categorisation enabled a balanced population in each group and made the results interpretable because primary school and below could be regarded as practically illiterate, and those that completed college and above could be seen as having higher education in China.

Four breastfeeding outcome variables, including early initiation of breast feeding (EIB), exclusive breast feeding (EBF) under 6 months, predominant breast feeding (PBF) under 6 months and children ever breast fed (ever BF), were analysed. EIB and ever BF were defined according to WHO indicators.[19] The final calculation of

EIB prevalence and ever BF prevalence was conducted among mothers with children from 0 to 12 months of age. EIB prevalence was defined as the proportion of children born in the last 12 months who were put to the breast within an hour after birth. Ever BF prevalence was defined as the proportion of children who were ever BF among all children aged 0–12 months. EBF and PBF were defined and evaluated according to Wellstart International's toolkit for monitoring and evaluating breastfeeding activities,[20] using a 24-hour recall methodology. Mothers were asked to recall the food they fed their children in the last 24 hours, and the final calculation of EBF prevalence and PBF prevalence was conducted among mothers with children from 0 to 6 months. EBF prevalence was defined as the proportion of infants 0–6 months of age who were exclusively fed breast milk. PBF prevalence was defined as the proportion of infants 0–6 months of age who were predominantly breast fed, which mainly comprised children who were fed by breast milk and water.[20] Furthermore, to explore mothers' current breastfeeding (CBF) situation, the present study also implemented an indicator of CBF prevalence, defined as any breast feeding in the last 24 hours of children less than 12 months of age.[20]

$$EBF = \frac{\text{children aged from } 0-12 \text{ months who were put to the breast within an hour after birth}}{\text{children aged from } 0-12 \text{ months}}$$

$$EBF = \frac{\text{children aged from } 0-6 \text{ months who were exclusively fed breast milk in the last 24 hours}}{\text{children aged from } 0-6 \text{ months}}$$

$$PBF = \frac{\text{children aged from } 0-6 \text{ months who were fed predominantly breast milk in the last 24 hours}}{\text{children aged from } 0-6 \text{ months}}$$

$$Ever\ BF = \frac{\text{children aged from } 0-12 \text{ months who were ever breastfed}}{\text{children aged from } 0-12 \text{ months}}$$

$$CBF = \frac{\text{children aged from } 0-12 \text{ months who were breastfed in the last 24 hours}}{\text{children aged from } 0-12 \text{ months}}$$

Mother–infant indicators and family socioeconomic status were the two categories of covariates incorporated in the present study. Mother–infant indicators included maternal age, infant age, pre-pregnancy body mass index (BMI), gestational age, infant birth weight, infant sex, parity and delivery method. All covariate data were collected from the mother's self-reported answers, aside from infant age and pre-pregnancy BMI. Infant age was calculated by the infants' birth date and survey date. Pre-pregnancy BMI was calculated based on the mothers' self-reported weight and height before pregnancy. Apart from maternal education, family socioeconomic status included household income, residency status and maternal occupation, which were also collected based on participants' self-reported answers. In the baseline survey, household income was a continuous variable. In the present study, household income was classified into ≤40 000 yuan (US$1 ≈ 6.33 Chinese yuan); 40 001–80 000 yuan and >80 000 yuan. Occupation was categorised as agriculture and related workers, factory workers, white-collar or others. Residency status was classified as local or migrant, according to whether the mother lives at their place of residence. Health-seeking behaviours, including considering breast feeding before pregnancy, attending postpartum mother support groups and receiving breast-feeding education during the perinatal period, were analysed in the present study.

### Data analysis

Descriptive analyses were used to illustrate the basic demographic, socioeconomic and lifestyle characteristics in different maternal education groups. The data from the mothers that did not complete the questionnaires as required were excluded from the multivariable analyses. Since the prevalence of the outcomes was high, the ORs tended to overestimate the risk ratios (RRs). Thus, a log-binomial (or binomial log-linear regression) model[21 22] was employed to calculate the RRs to explore the association between maternal education and breast-feeding outcomes. All of the RRs were presented with a 95% CI. The primary school and uneducated group were chosen to be the reference groups for all models. Two models were fitted: (1) unadjusted; (2) adjusted for maternal age, infant birth weight, infant sex, parity, delivery method, household income, region (super city/urban/rural/poor rural), residency status (local/migrant), paternal education and maternal occupation. The adjusted variables were chosen by reviewing relevant literature and was in accordance with the social determinants of breastfeeding framework.[6]

To further understand the relationship between maternal education and EIB and EBF, the adjusted prevalence of the three variables was calculated after stratification by household income. The adjusted prevalence of health-seeking behaviours, including considering breast feeding before pregnancy, attending a postpartum mother support group and receiving breastfeeding education, were also calculated in the different education groups. The adjusted prevalence was based on logistic regression, and the specific method is described elsewhere.[23] All the analyses were conducted using SAS V.9.4 (SAS Institute, Cary, NC, USA).

### Patient involvement

Patients were not involved in setting the research questions or planning the study. Investigators, including both the interviewers and the researchers, do not know the identities of the study participants.

## RESULTS

Table 1 shows the baseline characteristics of the study population by maternal education. Of all 10 408 participants that were included, 7.50% finished their education at primary school or below, 36.91% had a middle school

education, 19.12% had a high/vocational school education and 36.46% had a college education and above. The mean maternal ages for those whose highest education was primary school and below, middle school, high/vocational school and college and above were 28.76 (SD=6.68), 28.41 (SD=5.21), 28.26 (SD=4.78) and 30.36 (SD=4.51), respectively. The mean infant age (in days) for mothers whose highest education was primary school and below, middle school, high/vocational school and college and above were 186.50 (SD=106.82), 182.11 (SD=104.48), 178.29 (SD=105.82) and 173.10 (SD=105.52), respectively. The detailed distribution of infant age at interview by maternal education can be found in online supplementary file 1. The distributions of pre-pregnancy BMI, gestational age, infant birth weight and infant sex were similar across all maternal education groups. Women with an education level of primary school or below tended to be multiparous, had vaginal deliveries, lived in urban and poor rural areas, and had lower household incomes. Women who had primary school education or below were more likely to hold agricultural jobs (57.89%) and were less likely to be white-collar (11.77%). Women whose educational level was college and above were more likely to be migrants (46.35%) compared with women of other educational levels.

The overall EIB, CBF, ever BF, EBF and PBF rates were 71.80%, 86.98%, 97.47%, 15.40%, and 36.37%, respectively. The prevalence of EIB was 64.85%, 68.24%, 71.11%, 77.21% and the prevalence of EBF was 26.53%, 15.36%, 13.64%, and 14.06% for mothers whose highest education was primary school and below, middle school, high/vocational school and college and above, respectively. Detailed breastfeeding rates in mothers with different educational levels are found at the end of table 1.

Table 2 presents the relationship between breastfeeding practices and maternal education. No significant association was found between CBF and maternal education. Compared with the primary school and below group, other groups showed a greater RR of EIB and ever BF. Compared with the primary school or below group, other groups had a smaller RR of EBF and PBF. After adjusting for possible confounders, the significant association between maternal education and breastfeeding outcomes was only found in the EIB (RR=1.05, 95% CI: 1.01 to 1.11 for mothers with a middle school education; RR=1.10, 95% CI: 1.03 to 1.17 for mothers with a high school/vocational school education and RR=1.16, 95% CI: 1.10 to 1.23 for mothers with a college or above education) and EBF (RR=0.66, 95% CI: 0.53 to 0.81 for mothers with a middle school education; RR=0.63, 95% CI: 0.49 to 0.81 for mothers with a high school/vocational school education and RR=0.74, 95% CI: 0.65 to 0.83 for mothers with a college or above education). The detailed RRs and 95% CIs are found in table 2.

Figure 1 presents the adjusted prevalence of EIB in the different maternal education groups, stratified by household income. The association between maternal education and EIB was only significant in those with household income >80 000 yuan. In this group, a lower prevalence of EIB (44.95%; 95% CI: 40.65% to 49.25%) was observed in those with a primary school education or below compared with those with a college education or above (78.39%; 95% CI: 76.25% to 80.52%).

Figure 2 presents the adjusted prevalence of EBF in different maternal education groups, stratified by household income. There was no significant association between maternal education and EBF prevalence in those with household income of 40 001–80 000 yuan. For those with household income of ≤40 000 yuan and >80 000 yuan, there was a general negative association between maternal education and the incidence of EBF. For those with household income of ≤40 000 yuan, a higher prevalence of EBF was observed in those with a primary school education or below (36.54%; 95% CI: 29.93% to 43.15%) to compared with those with a college education or above (11.61%; 95% CI: 10.58% to 12.65%). For those with a household income of >80 000 yuan, a higher prevalence of EBF was observed in those with a primary or below education (26.83%; 95% CI: 26.63% to 27.03%) compared with those with a college education or above (9.69%; 95% CI: 8.14% to 11.23%).

Figure 3 presents the adjusted prevalence of health-seeking behaviours, including considering breast feeding before pregnancy, attending a mother support group postpartum and receiving breastfeeding education. There was a general positive association between health-seeking behaviours and maternal education across these three factors. A lower prevalence of considering breastfeeding before pregnancy was observed in those with a primary school education or below (28.35%; 95% CI: 25.98% to 30.73%) compared with those with a college education or above (53.92%; 95% CI: 52.09% to 55.75%). A higher prevalence of attending mother support groups postpartum was observed in those with a primary school education or below (10.09%; 95% CI: 9.20% to 10.98%) compared with those with a college education or above (56.03%; 95% CI: 54.55% to 57.51%). A lower prevalence of receiving breastfeeding education was also observed in those with a primary school education or below (32.65%; 95% CI: 30.71% to 34.59%) compared with those with a college education or above (66.47%; 95% CI: 65.16% to 67.78%).

## DISCUSSION

Several significant findings are illustrated in the present study. First, maternal education is inversely related to EBF. Second, maternal education is positively related to the EIB in the high-income households. Third, after stratification by income, the patterns of association between maternal education and breastfeeding indicators differed in the different income groups. The positive relationship between maternal education and EIB is observed in the group with the highest household income. An inverse relationship between maternal education and EBF exists in the low household and high household income

**Table 1** Demographic and social-economic characteristics of the participants

| | Maternal education | | | |
|---|---|---|---|---|
| | Primary school and below (N=781) | Middle school (N=3842) | High school/vocational school (N=1990) | College and above (N=3795) |
| Mean maternal age, year (SD) | 28.76 (6.68) | 28.41 (5.21) | 28.26 (4.78) | 30.36 (4.51) |
| Mean pre-pregnancy BMI, kg/m$^2$ (SD) | 22.67 (4.66) | 22.44 (4.16) | 21.74 (4.04) | 21.75 (4.12) |
| Mean gestational age, week (SD) | 39.02 (1.38) | 39.03 (1.40) | 38.94 (1.51) | 38.98 (1.38) |
| Mean infant birth weight, kg (SD) | 3.23 (0.47) | 3.33 (0.59) | 3.34 (0.67) | 3.43 (0.75) |
| Mean infant age, day (SD) | 186.50 (106.82) | 182.11 (104.48) | 178.29 (105.82) | 173.10 (105.52) |
| Infant sex, % | | | | |
| Male | 50.67 | 50.64 | 52.17 | 50.08 |
| Female | 49.33 | 49.36 | 47.83 | 49.92 |
| Parity, % | | | | |
| Primiparous | 29.70 | 33.87 | 43.49 | 62.06 |
| Multiparous | 70.30 | 66.13 | 56.51 | 37.94 |
| Delivery method, % | | | | |
| Vaginal | 65.77 | 56.72 | 54.59 | 57.01 |
| Caesarean section | 34.23 | 43.28 | 45.41 | 42.99 |
| Region, % | | | | |
| Super city | 9.70 | 14.16 | 30.82 | 61.55 |
| Urban | 43.15 | 32.08 | 39.09 | 27.41 |
| Rural | 7.15 | 25.39 | 21.09 | 7.29 |
| Poor rural | 40.00 | 28.37 | 9.00 | 3.75 |
| Household income, % | | | | |
| ≤40 000 yuan | 61.74 | 37.44 | 20.06 | 11.81 |
| 40 000–79,999 yuan | 22.39 | 30.01 | 26.21 | 13.82 |
| ≥80 000 yuan | 3.48 | 7.84 | 16.40 | 38.89 |
| Maternal occupation, % | | | | |
| Agriculture related | 57.89 | 33.61 | 9.37 | 2.27 |
| Factory workers | 11.53 | 19.95 | 20.31 | 8.66 |
| White-collar | 11.77 | 26.74 | 47.23 | 73.03 |
| Others* | 18.81 | 19.71 | 23.09 | 16.04 |
| Resident status, % | | | | |
| Local | 69.70 | 69.15 | 59.50 | 53.65 |
| Migrant | 30.30 | 30.85 | 40.50 | 46.35 |
| Breastfeeding practice, % | | | | |
| Early initiation | 64.85 | 68.24 | 71.11 | 77.21 |
| Current BF | 87.88 | 88.53 | 85.09 | 86.21 |
| Ever BF | 95.52 | 97.31 | 97.65 | 97.93 |
| Exclusive BF (0–6 months) | 26.53 | 15.36 | 13.64 | 14.06 |
| Predominant BF (0–6 months) | 50.00 | 39.99 | 18.44 | 39.29 |
| Health-seeking behaviours | | | | |
| Considered feeding method before pregnancy | 28.31 | 45.35 | 50.24 | 55.29 |
| Attended mother support group | 8.24 | 24.40 | 38.20 | 49.36 |
| Received breastfeeding education | 38.59 | 47.23 | 59.13 | 64.23 |

*Others refer to unidentified occupation, unemployed and housewives
BF, breast feeding; BMI, body mass index; ever BF, ever brest fed.

**Table 2** Relations between breastfeeding practices and maternal education*

| | Unadjusted risk ratio (95% CI) | | | |
| --- | --- | --- | --- | --- |
| | Primary school and below | Middle school | High school/vocational school | College and above |
| Early initiation | 1 | 1.05 (0.97 to 1.13) | 1.10 (1.01 to 1.19) | 1.19 (1.09 to 1.29) |
| Current BF | 1 | 1.01 (1.01 to 1.01) | 0.97 (0.96 to 1.00) | 0.98 (0.97 to 1.01) |
| Ever BF | 1 | 1.02 (1.02 to 1.03) | 1.02 (1.02 to 1.02) | 1.03 (1.03 to 1.03) |
| Exclusive BF (0–6 months) | 1 | 0.58 (0.52 to 0.64) | 0.51 (0.42 to 0.60) | 0.53 (0.47 to 0.59) |
| Predominant BF (0–6 months) | 1 | 0.80 (0.72 to 0.87) | 0.37 (0.24 to 0.50) | 0.79 (0.70 to 0.89) |
| | Adjusted risk ratio (95% CI) | | | |
| | Primary school and below | Middle school | High school/vocational school | College and above |
| Early initiation | 1 | 1.05 (1.01 to 1.11) | 1.10 (1.03 to 1.17) | 1.16 (1.10 to 1.23) |
| Current BF | 1 | 1.01 (0.99 to 1.04) | 1.02 (0.99 to 1.05) | 1.04 (1.01 to 1.07) |
| Ever BF | 1 | 1.01 (1.01 to 1.01) | 1.01 (1.01 to 1.01) | 1.01 (1.01 to 1.01) |
| Exclusive BF (0–6 months) | 1 | 0.66 (0.53 to 0.81) | 0.63 (0.49 to 0.81) | 0.74 (0.65 to 0.83) |
| Predominant BF (0–6 months) | 1 | 0.89 (0.79 to 1.00) | 0.84 (0.72 to 0.97) | 0.73 (0.62 to 0.85) |

*Adjusted for maternal age, infant birth weight, infant sex, parity, delivery method, household income, region (super city/urban/rural/poor rural), residency (local/migrant), paternal education and occupation.
BF, breastfeeding; ever BF, ever breastfed.

groups, and the association is stronger in the low-income households.

This study is based on a large, national representative, population-based survey. The large sample size ensures high precision in the findings. The associations observed in the present study accorded with and contributed to the development of the social determinants of breastfeeding framework.[6] This is one of the few studies mainly focused on the association between maternal education and breastfeeding practices in China. We specified breastfeeding indicators, explored the associations separately, and stratified using possible influential factors. Quantitative data were used to analyse the health-seeking behaviours. Admittedly, there are some potential limitations in the present study. First, causal inference between maternal education and breastfeeding practices is limited due to the study being cross-sectional. Further prospective study is needed to establish a causal relationship. Second, maternal education in this study represents formal education gained through schooling, and it may not reflect the actual health literacy of the mothers. Third, feeding practices were determined using the 'last-24-hour method', which tends to overestimate the prevalence of EBF as compared with the 'since birth method',[24] and the information used in the analysis was all based on mothers' self-report, which might introduce recall biases. Last, one of the potential limitations of the present study is that we did not survey the number of current births

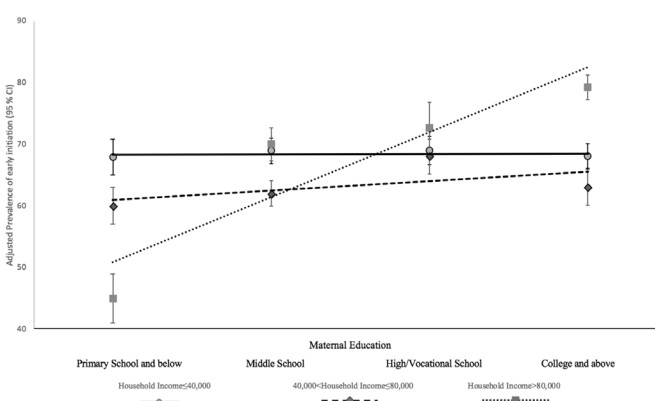

**Figure 1** Adjusted prevalence* of early initiation in different maternal education groups, stratified by household income. *After adjusting for maternal age, infant sex, parity, delivery method, paternal education, region (super city/urban/rural/poor rural), residency (local/migrant) and occupation.

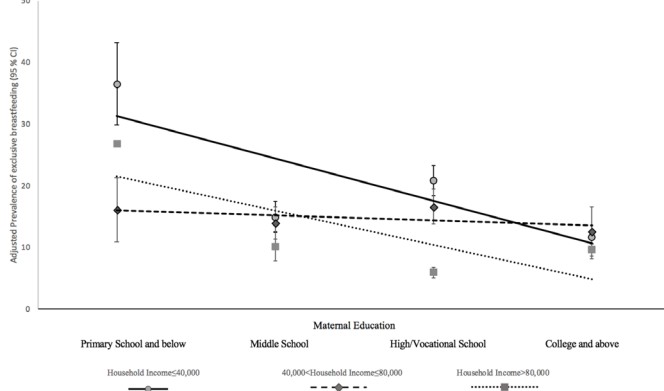

**Figure 2** Adjusted prevalence* of exclusive breast feeding in different maternal education groups, stratified by household income. *After adjusting for maternal age, infant sex, parity, delivery method, paternal education, region (super city/urban/rural/poor rural), residency (local/migrant) and occupation.

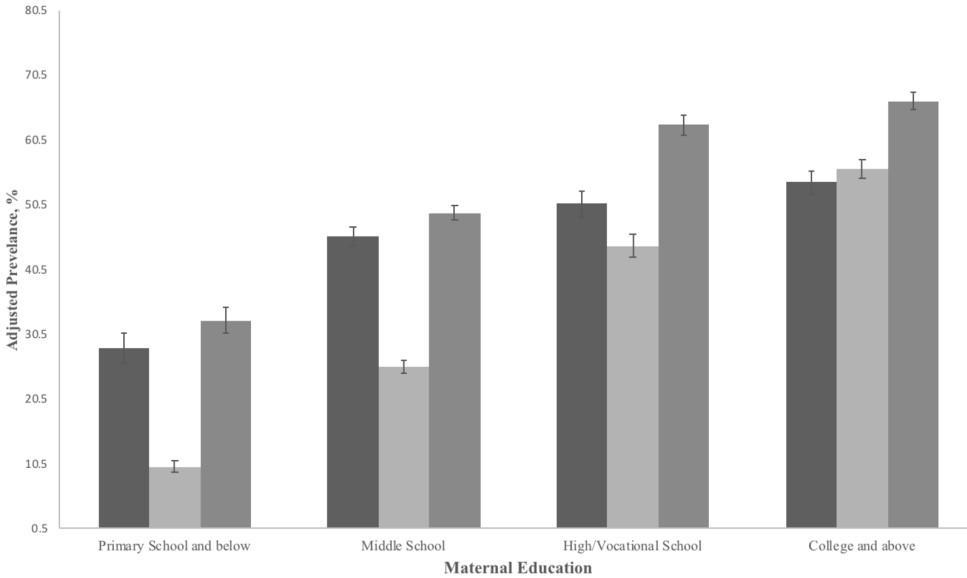

**Figure 3** Adjusted prevalence* of considering feeding method before pregnancy, attending mother support group, receiving breastfeeding education in different maternal education groups. *After adjusting for maternal age, infant sex, parity, delivery method, paternal education, household income, region (super city/urban/rural/poor rural), residency (local/migrant) and occupation.

among the mothers. As the association between maternal education and infant breastfeeding practices might differ in twins and singletons, this could affect the accuracy of the present findings.

In contrast to the EIB, maternal education is inversely related to EBF, which is consistent with previous studies conducted in China.[18] A systematic review conducted in China exploring the relationship between maternal education and breastfeeding indicators found a negative association. The review speculated that maternal leave, occupation and incorrect traditional perceptions contributed to the negative association.[18] One of the traditional perceptions was that breast feeding would change the mothers' body shape and have a negative impact on her health.[25] Nevertheless, our finding was opposite to those in developed countries.[14 15] In China, highly educated mothers are usually working full time in responsible and formal positions. The maternity leave regulated by legislation is 98 days in China,[26] which is not enough to guarantee the 6-month EBF time recommended by WHO. It is also equally difficult for mothers who have already returned to work to continue breast feeding due to the lack of related supporting facilities and increased working pressure.[27] Another possible explanation is that highly educated mothers tend to have higher household incomes, and thus are able to afford infant formula, which is especially relevant in China.[28] Due to prevalent advertising of infant formula and a lack of proper breastfeeding education, the use of infant formula is sometimes regarded as healthier and is also a symbol of wealth in China.[28 29] Besides, incorrect perceptions in some areas maintain that breastfeeding changes a mothers' body shape.[25] Women with higher education generally live in urban areas and are of a relatively higher social class.

Women in those urban areas and higher social classes may face higher social pressures. Thus, they may pay more attention to their body shape and are more willing to choose infant formula.[30]

In the present study, a positive association between the EIB and maternal education is observed in the household incomes >80 000 yuan. The positive association between maternal education and early initiation is consistent with studies conducted in the developed countries.[14 15 31] Previous studies in developing countries showed inconsistent results concerning the relationship between maternal education and the EIB.[13 32–34] A study conducted in Laos found that the EIB was most prevalent among mothers with higher education[32] and a study in Nepal observed the similar result.[13] A study conducted in Vietnam, however, found that mothers with no education were more likely to initiate breast feeding.[34] Results from the WHO Global Survey that was conducted primarily in developing countries found an overall negative association between maternal education and EIB.[33] The positive association in this study needs to be explored. In the present study, mothers with higher educational attainment tended to exhibit proactive health behaviours, early consideration of feeding methods and were more likely to receive breastfeeding education, all of which may foster the EIB. It has been suggested that women with higher education tend to have stronger social support,[35] and are more likely to attend mother support groups, which was also observed in this study. Previous studies have found that social support is an important contributor to the EIB.[36 37] Furthermore, women with higher education are more likely to live in super cities and have a higher household income. There is a considerable disparity of health services between rural and urban areas in China.[38] Thus,

women who live in super cities are more likely to have better access to quality health services. Previous studies found that the professional knowledge from nurses and doctors may foster EIB.[39] [40] A review conducted in India suggested that interventions designed to increase the knowledge and skills of healthcare workers were important in promoting the EIB.[41]

After stratification by income, the patterns of association between maternal education and breastfeeding indicators differed across income groups. The positive relationship between maternal education and the EIB was only significant in the high household income group. Higher education may promote better breastfeeding knowledge and health behaviours.[42] Nevertheless, apart from knowledge, the EIB requires social support and access to healthcare services.[3] Women with lower household incomes may not have the financial ability to obtain healthcare services. These women are more likely to give birth at home, where initiation of breast feeding within 1 hour of birth may not be promoted. Mothers may recognise the benefits of delivery at healthcare facilities, yet they may not be able to afford the expensive hospitals stays and professional services. The inverse relationship between maternal education and EBF was the strongest in low-income households. A possible explanation is that women with a lower income and higher education tend to have a heavier workload to meet the demands of the family, which consequently shortened the duration of EBF. People with a lower income tend to live in more rural areas, where health infrastructure and the expertise of healthcare workers may be less advanced than urban areas. Traditional values may also be more prevalent than in the cities, such as routine supplementary feeding with water,[43] which may have given rise to the associations found in this study.

Associations found in the present study had implications for future public health interventions to promote breast feeding in China. There are also several related factors to consider when designing such interventions. In the present study, the overall EBF rates were 15.40%. A national representative survey of 14 539 infants aged under 2 years in 55 counties from 30 provinces in China indicated that the crude EBF rates were 20.7% and weighed EBF rates were 18.6%.[44] The lower EBF rates observed in this research might indicate that EBF rates have been decreasing from 2013, which should raise the attention of policymakers and public health workers. In the present study, ever BF rates were much higher than EIB rates, which might be partly due to the high caesarean section prevalence in China.[7] Baby Friendly Hospital Initiative is utilised in most of the hospitals with obstetrical departments in China and has had positive impacts on breastfeeding practices, but the implementation of this Initiative was highly questionable due to a lack of routine monitoring and evaluation, which might reduce EIB rates.[7] [45] Sociocultural norms in China, such as postpartum confinement, are widely practised, which may also affect mothers' breastfeeding behaviours.[43] In addition, despite a nation-wide melamine milk scandal in 2008, the infant formula marketing became even more prevalent in China[28] [29] after the Chinese government abolished the International Code of Marketing of Breast-milk Substitutes.[46] All of these suggest that an extensive public campaign involving multisectoral stakeholders to promote breastfeeding practices is highly needed in China.

## CONCLUSIONS

A positive association between maternal education and the EIB and a negative association between maternal education and EBF were observed in this study, with the consideration of health-seeking behaviour and household income. Routine and successful breast feeding, as per the American Academy of Paediatrics recommendation,[1] is crucial for the health of the infant and is primarily influenced by social factors. This study underscores the influence of maternal education on breastfeeding practices. Findings from this study contribute to the growing body of research on the social determinants of breast feeding. This study suggests that extensive public health interventions on breastfeeding promotion in China should target mothers with higher levels of education, especially among those with medium to high incomes. Appropriate supportive policies and programme for this group of mothers, including enforcing full-term maternal leave, providing breastfeeding education and counselling, building a mother-friendly workplace, as well as breaking cultural and social norms, should be implemented to create an encouraging environment for breastfeeding practice among Chinese mothers.

**Contributors** KT and HW contributed to the study concept and design, statistical analysis, results interpretation, and drafting and revision of the manuscript. SHT contributed to the study concept and design, drafting and revision of the manuscript. TX contributed to the study concept and design, revision of the manuscript. YW and TT contributed to revision of the manuscript. YL and XQ contributed to the study concept and design and results interpretation. JG contributed to the study concept and design. All authors read and approved the final version of the manuscript.

**Funding** This study was supported by Bill & Melinda Gates Foundation and China Development Research Foundation.

**Competing interests** None declared.

**Patient consent for publication** Not required.

**Provenance and peer review** Not commissioned; externally peer reviewed.

**Data availability statement** Data are available upon reasonable request.

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
