## [Reviewer comments · BMJ Open]

ARTICLE DETAILS

TITLE (PROVISIONAL)	The association between maternal education and breastfeeding practices in China: a population-based cross-sectional study
AUTHORS	Tang, Kun; Wang, Hanyu; Tan, Shi Hui; Xin, Tong; Qu, Xueqi; Tang, Tianyu; Wang, Yuqi; Liu, Yuning; Gaoshan, Junjian

VERSION 1 – REVIEW

REVIEWER	Marina Cuttini, senior researcher Pediatric Hospital Bambino Gesù, Roma, Italy
REVIEW RETURNED	14-Jan-2019

GENERAL COMMENTS	This paper reports the results of a large cross-sectional study aiming at exploring the relation between maternal education and breastfeeding (BF) in China. Mothers whose children were <12 months of age were eligible for this study. The study used the following main outcomes: 1. early initiation of BF (EIF);2. exclusive BF under 6 months (EBF);3. predominant BF under 6 months (PBF);4. children ever breastfed (Ever BF). Additionally, current breastfeeding (CBF) was also collected. The strengths of this study are the large, representative sample and the use of the WHO definitions of BF. The paper is well written. Main concerns: 1) two of the outcome measures (EBF under 6 months and PBF under 6 months) plus current breastfeeding prevalence (CBF) were collected using the “last 24-hour” recall methodology. I feel this is not the best method to explore associations in an epidemiological study, where precise information at individual level is needed. Different child age distribution in mother with different education may affect validity of estimated associations. How did the Authors deal with this issue in data collection and/or analysis? It seems that in this study mother’s interviewing may have taken place at different times from birth, is this correct? If not, please state child’s age at interview.
---

	2) Logistic regression models were used to compute adjusted ORs and 95% confidence intervals. However, ORs are not the best measure when the outcome is very frequent (as it is the case with BF in this study); modeling of Risk Ratios rather than Odds Ratios would be more appropriate. Additional comments: Introduction  - in third paragraph, the Authors state that findings about the relation between maternal education and BF are inconsistent. Although I did not carry out any systematic review I feel that , apart from very old reports, most studies show that higher education favours BF. It might be different in China (see ref. 16), but as regards the studies quoted in this paragraph to support claims of inconsistency: a) Ref 11, 12 and 13 support the association between higher maternal education and BF; a) Ref. 14 does not deal with BF (only infant mortality, child height and immunizations); b) Ref 15 (Skafida 2009) appears to be misunderstood by the Authors. Skafida aimed at assessing whether maternal education or occupation-based social class better predicted BF take-up. She concluded that “compared with occupation-related social class, maternal education is more informative, accurate and useful lens through which to understand and explain patterns of breastfeeding take-up”. I suggest that Authors modify their statement by saying, for instance, that while most studies report a positive association between maternal education and BF, this may not be true in all countries, and provide the appropriate references. Methods:  - Study design and participants, first paragraph: could Authors provide some more information about data sources and variables used to take into account the “population structure and status of maternal and child health and socio-economic development” in population proportionate sampling? - when did data collection take place for this study? - Variable definitions, last paragraph, “according to whether the mother lives at the place of domicile”: should place of residence be used instead of domicile? Data analysis:  -in Figures footnotes (*), also paternal education is mentioned as adjustment variable; however, it is not the list quoted in text, nor in Table 2 footnote. Please explain the reasons for this inconsistency. - also, was the information about multiple birth collected? If yes, it should be taken into account, as sustained BF in twins may be more difficult than for singletons. - Can health-seeking behaviours be conceptualized as Mediators of the association between maternal education and BF (EIF and Ever BF)? Did Authors consider carrying out a mediator analysis? Discussion  - Fourth paragraph, page 12. In second row, Authors refer to “This result is consistent with...”: what do they mean by “This result”? The positive association between education and EIF, or the fact that this association is observed only among high income women?
--	---

	Additionally they add that “this result” is not observed in studies from developing countries. In support however they quote papers (Ref 28 and 29) considering exclusive breastfeeding (and not early initiation). As different BF indicators may have different meaning and associations, it would be better not to use them interchangeably, unless such use is explicitly motivated. Figures I suggest not to use connecting lines between the adjusted ORs, as maternal education is measured on a categorical scale.
--	--

VERSION 1 – AUTHOR RESPONSE

Responses to Reviewer 1

Marina Cuttini, senior researcher

Institution and Country: Pediatric Hospital Bambino Gesù, Roma, Italy

Point by point response:

1. For Ethics, I could not find anything about Informed Consent by study participants.

All the participants in this study provided informed consent. We added the following statement at the end of Study design and participants part:

“All the participants included in this study provided informed consent.” (See Page 6, Line 40-43)

2. As regards outcome definition, I could not find the child's age at interview.

As the reviewer suggested, we agree that it is important to add infant age in our study because it was an important attribute of the study subjects.

Thus, infant age was added in the Variable definitions part as “Infant age was calculated by infants’ birth date and survey date.” (See Page 8 Line 11-12)

Subsequently, the main infant age was presented in the updated Table 1 and the results part also described the following information: “The mean infant age (in days) for mothers whose highest education was primary school and below, middle school, high/vocational school, and college and above were 186.50 (SD=106.82), 182.11 (SD=104.48), 178.29 (SD=105.82), and 173.10 (SD=105.52), respectively.” (Page 9 Line 60 – Page 10 Line 4)

Also, relevant analysis that examined the modifying effects of infant age was conducted.

3. I think it would be better to focus the analyses on early initiation and ever breastfed children only.

We do, to a certain extent, agree with the reviewer on this point because only those who ever breastfeed their baby can exclusively feed their baby. However, the study was conducted in the whole survey population for the following reasons:

- 1) First and foremost, the statistical reliability and interpretations of results were based on a three-stage population representative random sampling. With sufficient statistical power, we do want to analyze all the three important breastfeeding indicators of exclusive breastfeeding, early initiation and ever breastfeeding in the study population. Our hypothesis was that maternal education would have major impact on all three breastfeeding indicators in this particular population
- 2) In our analysis, we are using the total surveyed population as the basis to calculate risk ratios. If confining the analysis to only ever-breastfeeding children, we believe we will then be answering a slightly different research question, which is, the association of maternal education level with EBF/Ever BF/Early Initiation among ever-breastfeeding mothers.

Main concerns:

1. Two of the outcome measures (EBF under 6 months and PBF under 6 months) plus current breastfeeding prevalence (CBF) were collected using the “last 24-hour” recall methodology.

I feel this is not the best method to explore associations in an epidemiological study, where precise information at individual level is needed.

We agree that this is certainly an important concern because it relates to our main outcomes and thus directly affects the study results. This was a major issue we took into consideration when the study was designed and the analysis was conducted. Measuring exclusive breastfeeding is a complex issue as rates can vary according to the different definition and measurement. The most frequently used methods are 24-h recall and recall since-birth.¹ We therefore discussed in the limitation part of the manuscript, acknowledging that feeding practices were determined using the 'last-24-hour method', which tends to overestimate the prevalence of EBF as compared to the 'since birth method'.¹ Recall since-birth method, however, can also be influenced by recall error.²

World Health Organization recommended the 'last-24-hour method' because this indicator represented the best option for estimating exclusive breastfeeding and was more sensitive to capturing changes.³ The 'last-24-hour method' was also implemented in some large scale researches, such as The Demographic and Health Surveys conducted in over 90 developing countries.⁴

Thus, we believe it is justifiable that we employed the 'last-24-hour method' while adding "feeding practices were determined using the 'last-24-hour method' which tends to overestimate the prevalence of EBF as compared to the 'since birth method' ." in the limitation part of the manuscript. Please refer to Page 12 Line 55 – 60

2. Different child age distribution in mother with different education may affect validity of estimated associations. How did the Authors deal with this issue in data collection and/or analysis? It seems that in this study mother's interviewing may have taken place at different times from birth, is this correct? If not, please state child's age at interview.

Infant age was indeed different in the study subjects and infant age was calculated by infant birth date and survey date.

As we mentioned before, we do agree with the reviewer that infant age was certainly an important factor to influence breastfeeding practices. We maintained, however, infant age was not a confounder for the association between maternal education and breastfeeding practice. Although infant age may affect breastfeeding practices, however, infant age (i.e. month since birth, all our interviewees were mothers whose child's age ≤ 1 years) was not directly related to maternal education, as illustrated below:

Infant age is not related to maternal education, which means that infant age is not a confounder in the association between maternal education and breastfeeding practices.

Nevertheless, we believe infant age was an important attribute and a useful indicator. Thus, we presented the mean infant age in Table 1.

Meanwhile, although infant age was not a confounder, it could well be an effect modifier. Thus, we conducted a sensitivity analysis and added infant age in the new adjusted model (See Attached Table 1 for Reviewers).

We found that by adjusting for infant age, maternal education was still positively related to EIB while negatively related to EBF and the strength of associations did not change significantly. In conclusion, considering infant age was not a mediator and did not substantially affect our study results, we therefore didn't include it in our main analysis.

Attached Table 1 for Reviewers

*Model 1 adjusted for maternal age, infant birth weight, infant sex, parity, delivery method, household income, region (super city/urban/rural/poor rural), residency (local/migrant), paternal education, and occupation

**Model 2 adjusted for infant age, maternal age, infant birth weight, infant sex, parity, delivery method, household income, region (super city/urban/rural/poor rural), residency (local/migrant), paternal education, and occupation

Logistic regression models were used to compute adjusted ORs and 95% confidence intervals.

	Model 1*: Corrected Adjusted Risk Ratio (95% CI)			
	Primary school and below	Middle School	High School/Vocational School	College and above
Early initiation	1	1.14(1.06,1.22)	1.16(1.06,1.25)	1.22(1.12,1.30)
Exclusive BF (0-6 months)	1	0.53(0.38,0.73)	0.66(0.44,0.96)	0.59(0.38,0.88)
	Model 2**: Corrected Adjusted Risk Ratio (95% CI)			
	Primary school and below	Middle School	High School/Vocational School	College and above
Early initiation	1	1.06(0.97,1.14)	1.12(1.02,1.21)	1.15(1.05,1.23)
Exclusive BF (0-6 months)	1	0.58(0.40,0.79)	0.65(0.42,0.96)	0.70(0.44,1.06)

However, ORs are not the best measure when the outcome is very frequent (as it is the case with BF in this study); modeling of Risk Ratios rather than Odds Ratios would be more appropriate. We agree with the reviewer that if a study outcome is common, the odds ratio will be further from 1 than the risk ratio.⁵ There has been debates about when to use odds ratio or risk ratio when conducting analysis, interpreting results and estimating associations in epidemiological studies.⁵ The overall exclusive breastfeeding rates was 15.40%, which was not very high. Nevertheless, overall EIB rate was 71.80%. Thus, it was better to calculate the risk ratio than just present odds ratio. We established corrected risk ratio, using the existed odds ratio, by employing the widely-used methods proposed by Zhang and Yu.⁶ We added the following statement: “Because the prevalence of the outcomes was high, odds ratio tended to overestimate the risk ratio. Thus, corrected risk ratio was calculated by the widely-used methods proposed by Zhang and Yu.⁶” in the Data analysis part (See Page 8 Line 50-56) and modified our results because of these changes. We updated Table 2 in the main text using Risk Ratios.

Additional comments:

Introduction

In third paragraph, the Authors state that findings about the relation between maternal education and BF are inconsistent. Although I did not carry out any systematic review I feel that , apart from very old reports, most studies show that higher education favours BF. It might be different in China (see ref. 16), but as regards the studies quoted in this paragraph to support claims of inconsistency:

- a) Ref 11, 12 and 13 support the association between higher maternal education and BF; a) Ref. 14 does not deal with BF (only infant mortality, child height and immunizations);
- b) Ref 15 (Skafida 2009) appears to be misunderstood by the Authors. Skafida aimed at assessing whether maternal education or occupation-based social class better predicted BF take-up. She concluded that “compared with occupation-related social class, maternal education is more informative, accurate and useful lens through which to understand and explain patterns of breastfeeding take-up”.

I suggest that Authors modify their statement by saying, for instance, that while most studies report a positive association between maternal education and BF, this may not be true in all countries, and provide the appropriate references.

Thanks you very much for this comment and thank you for the insights into the different references we used in this paper. There was a mistake concerning Ref 15 in the original manuscript because, as the reviewer suggested, we did not interpret the reference properly. We modified this paragraph to interpret the previous literature more accurately. Specifically, we moved the Ref 14 to the first sentence: "Education status of the mother has been identified as an important determinant for child welfare, survival, and health."(See Page 4 Line 50-53), because Ref 14 indicated that maternal education is related to infant mortality, child height and immunizations. Skafida's study was reinterpreted as:

"Skafida et al found that maternal education was a more informative predictor of breastfeeding initiation than occupation."(See Page 4 Line 53-56) We added studies from Ethiopia and Bangladesh where they found a negative association between maternal education and breastfeeding practices and modified this section as suggested: "Although maternal education and breastfeeding practices have been studied extensively and most of the previous literature reported a positive association between maternal education and breastfeeding practices, the positive association may not be true in all context. Studies conducted in Nepal, United States, and Italy showed that higher maternal education was related to better breastfeeding practices. while studies in Ethiopia and Bangladesh observed a negative association of breastfeeding and maternal education." (See Page 4 Line 55 – Page 5 Line 7)

Methods:

- Study design and participants, first paragraph: could Authors provide some more information about data sources and variables used to take into account the "population structure and status of maternal and child health and socio-economic development" in population proportionate sampling?

The survey was designed by the China CDC and they took a set of socio-economic, geographical, and demographic factors into consideration. We added the following statement in the study design and participants section:

"The selection of 12 study sites was made carefully, aiming to maximize geographic diversity (including northern, southern, eastern and western regions with very different climate and cultures), socioeconomic diversity (including affluent cities and impoverished inland rural areas, as well as cities of different population sizes) and taking into account past maternal and child health situation (i.e. the maternal mortality rates), immigration population proportion, population stability, local commitment and capacity in order to provide a balanced and representative sampling."(See Page 5 Line 42 - 56)

- when did data collection take place for this study?

Thank you for your kind reminder.

"This population-based study was conducted in 12 geographically defined regions in China from July 2017 to January 2018." was added to the Study design and participants part.(See Page 5 Line 34 - 38)

- Variable definitions, last paragraph, "according to whether the mother lives at the place of domicile": should place of residence be used instead of domicile?

Revised as required. (See Page 8 Line 29-30)

Data analysis:

-in Figures footnotes (*), also paternal education is mentioned as adjustment variable; however, it is not the list quoted in text, nor in Table 2 footnote. Please explain the reasons for this inconsistency. Sorry for making such a mistake in the manuscript. We did mean to put paternal education in our analysis in all our adjusted models because paternal education was considered as a confounder in the association. We checked our code and verified that we did adjust for paternal education in the adjusted models in Table 2 and we added the paternal education in the data analysis part (See Page 9 Line 1-4) and in the footnote of Table 2.

- also, was the information about multiple birth collected? If yes, it should be taken into account, as sustained BF in twins may be more difficult than for singletons.

We do agree with the reviewer that multiple birth could affect breastfeeding practices.

A study conducted in Sweden found that mothers with lower education were more likely to were subject to earlier cessation of breastfeeding⁷, whereas studies conducted among singletons in developed countries generally found a positive association between breastfeeding and education.^{8,9}

	Model 1*: Adjusted OR (95% CI)			
	Primary school and below	Middle School	High School/Vocational School	College and above
Early initiation	1	1.14(1.06,1.22)	1.16(1.06,1.25)	1.22(1.12,1.30)
Exclusive BF (0-6 months)	1	0.53(0.38,0.73)	0.66(0.44,0.96)	0.59(0.38,0.88)
	Model 2**: Model 1 + health seeking behaviors-Adjusted OR (95% CI)			
	Primary school and below	Middle School	High School/Vocational School	College and above
Early initiation	1	1.06(0.97,1.14)	1.12(1.02,1.20)	1.14(1.04,1.23)
Exclusive BF (0-6 months)	1	0.62(0.42,0.90)	0.76(0.48,1.17)	0.65(0.37,1.09)

Thus, it is reasonable to assume that the association between maternal education and infant breastfeeding practices differs in twins and singletons, which may affect the result of the present study. The present survey, however, did not collect relevant data. Thus, we discussed this in the limitation part of the study as: "One of the potential limitations of the present study is that we didn't survey the number of current births among the mothers. As the association between maternal education and infant breastfeeding practices might differ in twins and singletons, this could affect the accuracy of the present findings."(Please see Page 13 Line 1 - 9)

- Can health-seeking behaviours be conceptualized as Mediators of the association between maternal education and BF (EIF and Ever BF)? Did Authors consider carrying out a mediator analysis?

Health-seeking behaviors were indeed considered as mediators of the association between maternal education and EIB/EBF. As Figure 3 indicated, health-seeking behaviors were positively associated with maternal education. Health-seeking behaviors, self-evidently, contributed to better breastfeeding practices. As the Attached Table 2 below demonstrated, both the negative association between maternal education and EBF and the positive association between maternal education and EIB were attenuated after furthering adjustment of health-seeking behaviors.

Thus, we considered health-seeking behaviors modify the association between maternal education and breastfeeding outcomes. We interpreted in the manuscript as: "The positive association in this study needs to be explored. In the present study, mothers with have higher educational attainment tend to exhibit proactive health behaviors, early consideration of feeding methods and more likely to receive breastfeeding education, all of which may foster early initiation of breastfeeding."(See Page 14 Line

24 - 33).

Attach Table 2 for Reviewers

*Model 1 adjusted for maternal age, infant birth weight, infant sex, parity, delivery method, household income, region (super city/urban/rural/poor rural), residency (local/migrant), paternal education, and occupation

**Model 2 adjusted for maternal age, infant birth weight, infant sex, parity, delivery method, household income, region (super city/urban/rural/poor rural), residency (local/migrant), paternal education, occupation, considering feeding method before pregnancy, attending mother support group, receiving breastfeeding education.

Discussion

- Fourth paragraph, page 12. In second row, Authors refer to “This result is consistent with...”: what do they mean by “This result”? The positive association between education and EIF, or the fact that this association is observed only among high income women?

Here we intended to discuss the positive association between maternal education and early initiation, because the mediating effects of household income were discussed later in a separate paragraph.

We modified the wording as: “The positive association between maternal education and early initiation is consistent with studies conducted in the developed countries.”(See Page 14 Line 6- 9)

Additionally they add that “this result” is not observed in studies from developing countries. In support however they quote papers (Ref 28 and 29) considering exclusive breastfeeding (and not early initiation). As different BF indicators may have different meaning and associations, it would be better not to use them interchangeably, unless such use is explicitly motivated.

We do agree with the reviewer that EBF and EIB were distinct outcomes and might have different associations with maternal education, as we found in the present research. Our rationale to put these citations here before was that EBF and EIB were both breastfeeding practices and thus it was reasonable to quote them here, but we could also see our logical flaws. Thus, we rewrite this part, quoting citations concerning EIB instead of EBF, and we added references from countries bordered China (Laos¹⁰ and Vietnam¹¹) and also from a WHO Global survey conducted mostly in developing countries.¹² The modified part was as following:

“The positive association between maternal education and early initiation is consistent with studies conducted in the developed countries.^{8,9,13} Previous studies in developing countries showed inconsistent results concerning the relationship between maternal education and early initiation of breastfeeding.^{10-12,14} A study conducted in Laos found that early initiation of breastfeeding was most prevalent among mothers with higher education¹⁰ and a study in Nepal observed the similar result.¹⁴ A Study conducted in Vietnam, however, found that mothers with no education were more likely to initiate breastfeeding.¹¹ Results from the WHO Global Survey, which was conducted mostly in developing countries, found an overall negative association between maternal education and early initiation of breastfeeding.¹²”(See Page 14 Line 6 – 25)

Figures

I suggest not to use connecting lines between the adjusted ORs, as maternal education is measured on a categorical scale.

The smooth lines were changed to Line of Best Fit estimated using Least Square Methods in Figure 1 and Figure 2.

Figure 1 Adjusted Prevalence* of early initiation in different maternal education groups, stratified by household Income

*After adjusting for maternal age, infant sex, parity, delivery method, paternal education, region (super city/urban/rural/poor rural), residency (local/migrant), occupation

Figure 2 Adjusted Prevalence* of exclusive breastfeeding in different maternal education groups, stratified by household Income

*After adjusting for maternal age, infant sex, parity, delivery method, paternal education, region (super city/urban/rural/poor rural), residency (local/migrant), occupation

Reviewer: 2

Reviewer Name: Seema Miharshahi

Institution and Country: School of Public Health, University of Sydney

I am concerned that breastfeeding may be more strongly associated with income rather than education as high income people with low education are less likely to initiate, and low income people with low education are more likely to exclusively breastfeed so it would be good for the authors to comment on this.

We strongly agree with the reviewer that household income plays a significant role in mediating the association between maternal education and breastfeeding outcomes. This was a major concern when we conducted the analyses. Household income was adjusted in all models, presented in Table 2 as well as in the figures. After adjusting for household income and other confounders, significant associations were still observed, which means that holding household income as a constant, maternal education was still associated with EIB and EBF. Thus, we believe the inverse association of maternal education with exclusive breastfeeding and the positive association with early initiation were robust findings. Nevertheless, as the reviewer suggested, household income is an important factor to be considered, so we did a stratified analysis to understand how income may modify the association between maternal education, and breastfeeding outcomes (See Figure 1 and Figure 2). Different shapes of associations between maternal education and breastfeeding outcomes were observed after stratification and we discussed and interpreted the different association at the fifth paragraph in the discussion part. (See Page 14 Line 60 – Page 15 Line 35)

There is also very little discussion of the context and cultural norms that may affect breastfeeding in China eg confinement for a month after birth, is confinement still practiced widely? – is that data collected?

There needs to be a stronger discussion of the context that may affect breastfeeding – social norms, are policies like the Baby Friendly Hospital Initiative practiced in China? These additions to the discussion will help with implications for planning programs for populations in China.

The reviewer was right to point out that cultural norms, such as confinement, were prevalent in China. Also, Baby Friendly Hospital Initiative had large impacts on the breastfeeding practices. With this national survey, we did a series of research papers regarding different aspects of the breastfeeding practices in China, and as the reviewer suggested, cultural norms and baby friendly hospital initiative are among the manuscripts which are currently being drafted. The present study, however, focused on the association between maternal education and we only discussed cultural norms and health system when relevant, such as: “Women in urban areas and at higher social classes may face a higher social pressure. Thus, they may pay more attention to their own body shape and are more willing to choose infant formula.”(See Page 13 Line 55 – 60) and “There exists a huge disparity of health services between rural and urban areas in China. Thus, the women who live in super cities are more likely to have better access to quality health services.”(Page 14 Line 42 - 48) and “Traditional values may also be more prevalent than in the cities, such as routine supplementary feeding with water,¹⁵ giving rise to the association found in this study.”(Page 15 Line 32 - 35) In the original version of the manuscript.

Along with this quantitative study, a qualitative study of Chinese mothers’ breastfeeding practices were conducted. Our team is also analyzing the qualitative data to investigate on how socio-cultural norms in China affecting breastfeeding practices. Health system was also a research topic being explored in-depth by our team. Hopefully, these research studies would be published soon and the future interventions could refer to the present research to consider how maternal education influence breastfeeding and other research for socio-culture and health system factors.

After considering the reviewer’s comments, we also thought it was necessary to mention those cultural norms and social context in our discussion. Thus, a paragraph was added in the end of the Discussion section:

“Associations found in the present study had implications for future public health interventions to promote breastfeeding in China. There are also several related factors to consider when designing such interventions. In the present study, Ever BF rates were much higher than EIB rates, which might partly due to the high caesarean section prevalence in China.¹⁶ Baby Friendly Hospital Initiative covered most of the hospitals with obstetrical department in China and have had positive impacts on breastfeeding practices, but the implementation of this Initiative was highly questionable due to a lack of routine monitoring and evaluation, which might reduce EIB rates.^{16 17} Sociocultural norms in China, such as postpartum confinement, were widely practiced, which might also affect mothers’ breastfeeding behaviors.¹⁵ In addition, despite a nation-wide melamine milk scandal in 2008, the infant formula marketing was even more prevalent in China^{18 19}, after the Chinese government abolished the International Code of Marketing of Breast-milk Substitutes.²⁰ All of these suggest that a large public campaign involving multisectoral stakeholders to promote breastfeeding practices is highly needed in China. . ”(See Page 15 Line 40 – Page 16 Line 7)

The manuscript is mostly well written, however there needs to be a few modifications to the English that will make the manuscript clearer I believe.

Abstract

Line 24-25 please clarify they are mothers of children aged <12mo

This information was added in the participants part as: “10,408 mothers with children from 0-12 months of age....” (See Page 2 Line 24-25)

Please add effect sizes to the abstract

We modified our abstract by adding the effect sizes as:

“A higher level of maternal education is positively associated with early initiation of breastfeeding (RR: 1.22, 95% CI: 1.12-1.30 for the highest education group) and was inversely associated with exclusive breastfeeding (RR: 0.59, 95% CI: 0.38-0.88 for the highest education group). Stratified by household income, a positive association with early initiation was observed only in the group with the highest household income (adjusted prevalence: 44.95%, 95% CI: 40.65%-49.25% in the lowest education group versus 78.39%, 95% CI: 76.25%-80.52% in the highest education group), and an inverse association with exclusive breastfeeding was found in both low and high household income groups.” (See Page 2 Line 50-Page 3 Line 4)

Introduction

It would be great to add to the discussion whether the analyses was based on a theoretical model of breastfeeding for example on page 5 line 10-12 social determinants of breastfeeding –

Our analyses adopted the social determinants of breastfeeding framework proposed by The Lancet.²¹. Thus, we added: “The adjusted variables were chosen by reviewing relevant literature and according to the social determinants of breastfeeding framework.²¹” (See Page 9 Line 3 - 7) in the Data analysis part and “The associations observed in the present study accorded with and contributed to the development of social determinants of breastfeeding framework.²¹” in the Discussion part (See Page 12 Line 34 - 38).

Page 5 line 42 spelling mistake ‘underdeveloped’

Thank you very much for your kind reminder. Correction was made for this spelling mistake. (Page 6 Line 6-7)

Page 5 line 53 should read ‘field experience’

The grammar error was corrected. (Page 6 Line 6)

Methods

Variable definitions of education – age 6 lines 19-30 the categories of education are not clear, please rewrite to mention the number of categories and why they were chosen.

Our previous description of the categories, such as “the first three categories were combined as primary school and below and the last three categories were combined as college and above.” may, as the reviewer suggested, caused some confusions. Also, we should have mentioned the reason why we combined these categories. Thus, we modified the part as : “In the present study, those with no formal school education, dropped out from primary school, or those with primary school education only were classified into the ‘Primary School and Below’ group; those with middle school education were classified as the ‘Middle School’ group; those with high/vocational school education were classified as the ‘High/Vocational School’ group; and those with an education of associate/junior college, university, postgraduate and above were classified as the ‘College and Above’ group. This categorization enabled balanced population in each group and made the results interpretable because primary school and below could be regarded as practically illiterate and those with college and above education could be seen as having high education in China.” in the variable definition part. (See Page 6 Line 55 – Page 7 Line 15)

Page 6 line 55 – how many 24hr recalls were undertaken for each subject? Was it reflective of usual intake – was it done on weekend/ weekdays/during festivals? What methods of standardisation was carried out

We implemented the 24-hour recall methodology using Wellstart International’s toolkit²² for monitoring and evaluating breastfeeding activities, as cited in our paper. 24-hour recall data of all liquids and solids consumed by the infants. Respondents was probed about the different kinds of liquids the infant may have received, including water, juice, milk, formula, and other liquids.

We agree with the reviewer that in an individual level the timing of the question might cause recall bias. This cross-sectional survey was conducted July 2017 to January 2018, which covered weekdays and weekends and the days of survey were essentially randomly picked. This study was a population-

representative study covering over 10,000 mothers. Given the large sample size and the random nature of the recall day, we considered that the biases suggested may cause a non-differential misclassification, which may not severely distort the associations observed in the present study.

Page 7 line 22 it is unclear whether was birthweight self-reported by mothers? Was it recorded by the place of birth eg in a discharge summary? How accurate would this as well as self-report of pre-pregnancy BMI be?

Thank you for this concern. Birthweight is an important indicator in China, which all hospitals reported to both the delivering mother and their relatives on a routine basis. Culturally speaking, the Chinese parents believe that higher birthweight suggested a healthier child. Therefore, it is very common for a mother to remember her child's birthweight. Although there might be recall bias, we consider the impact to be minimal. We added this limitation in the Discussion part as: "...and the information used in the analysis was all based on mothers' self-report, which might introduce recall biases." (See Page 12 Line 60 – Page 13 Line 1)

Page 8 line 43-48 Patient involvement section seems quite random – they were not patients? Please consider whether this section should be removed.
As we understood, this section was required by the journal.

Results

Page 9 please specify what 'migrant residents' means

We are sorry that this wording was probably a little bit confusing. 'migrant residents' was changed into 'migrants' in this particular context. (See Page 10 Line 19-20) We defined this in the variable definition part of the manuscript as: "Residency status was classified as local or migrant, according to whether the mother lives at the place of residence." (See Page 8 Line 27-30)

In the results it would be good to report the overall prevalences of the five different outcomes of breastfeeding (this could go before the section on table 2)

The overall breastfeeding rates were added before the section on Table 2 as: "The overall EIB, CBF, Ever BF, EBF, and PBF rates were 71.80%, 86.98%, 97.47%, 15.40%, and 36.37%, respectively. Detailed breastfeeding rates in mothers with different educational levels could be found at the end of Table 1." (Page 10 Line 24 -30)

The current breastfeeding rate seems very high >80% it would be helpful to know the average age of the children.

Infant age was added in the Variable definitions part as "Infant age was calculated by infants' birth date and survey date." (See Page 8 Line 11-12)

The main infant age was presented in updated Table 1 and the results part also described the added information: "The mean infant age (in days) for mothers whose highest education was primary school and below, middle school, high/vocational school, and college and above were 186.50 (SD=106.82), 182.11 (SD=104.48), 178.29 (SD=105.82), and 173.10 (SD=105.52), respectively." (Page 9 Line 60 – Page 10 Line 4)

It also seems that ever breastfeeding rate was very high and initiation (within 1 hour of birth) was much lower, why is this – this should be added to the discussion.

Ever breastfeeding rate, as the review pointed out, was higher than EIB rate, which was a fact needs to be paid attention to.

Ever BF prevalence was defined as the proportion of children born in the last 12 months who were ever breastfed their child. EIB prevalence was defined as the proportion of mothers with children born in the last 12 months who were put to the breast within an hour after birth. In other words, those who

initiated breastfeeding within an hour after birth must have also ever breastfed their baby. Thus, logically EIB should be lower than Ever BF. Previous research, as summarized by Victora et al in The Lancet found that early initiation was low in all settings (average below 60% in low income countries and below 40% in low-middle income countries) while ever BF was over 80%, mostly over 90% percent in all countries.²³ The present study observed an overall EIB rates of 71.80% and Ever BF rates of 97.47%, which means that the difference did not deviate from previous findings.

We do agree, however, that this observation should be discussed. We added: “In the present study, Ever BF rates were much higher than EIB rates, which might partly due to the high caesarean section prevalence in China.¹⁶ Baby Friendly Hospital Initiative covered most of the hospitals with obstetrical department in China and have had positive impacts on breastfeeding practices, but the implementation of this Initiative was highly questionable due to a lack of routine monitoring and evaluation, which might reduce EIB rates.^{16 17}” in the Discussion part. (See Page 15 Line 45 -56)

Please remove causal language eg page 9 line 55 “increase” is incorrect as this is a cross-sectional study the correct term is ‘higher’ prevalence

Thank you for pointing this out, we modified all the wording as required in this part to avoid implying a causal relationship. (Please see Page 10 - 11)

Please remove all similar causal language through results especially page 10 lines 32-56

We modified the wording as required in this part. (Please see Page 10 - 11)

Please provide a statement around women who didn’t consent to participate in the study – were they similar in terms of the demographics in Table 1 ?

The reviewer was right to point out that the non-response population might cause potential inaccuracies for the findings. We couldn’t, however, to access the information of those who did not respond to the survey or those who refused to participate after the informed consent process. It was therefore not possible to present their characteristics. Because the response rate was over 90%, which was a very good response rate for a large scale national representative study, we consider the possibility of potential non-response biases was minimal.

See Figure 1 and 2 I am concerned that breastfeeding may be more strongly associated with income rather than education as high income people with low education are less likely to initiate, and low income people with low education are more likely to exclusively breastfeed so it would be good for the authors to comment on this.

Please refer to our response to the same question raised by the other reviewer above.

Discussion

The discussion is very well written but please add self-report to limitations section

We add this limitation as: “...and the information used in the analysis was all based on mothers’ self-report , which might include recall biases.” (See Page 12 Line 60 – Page 13 Line 1)

as well as a discussion on why ever breastfeeding rate was very high and initiation (within 1 hour of birth) was much lower

Please refer to our response above. In brief, we added: “In the present study, Ever BF rates were much higher than EIB rates, which might partly due to the high caesarean section prevalence in China.¹⁶ Baby Friendly Hospital Initiative covered most of the hospitals with obstetrical department in China and have had positive impacts on breastfeeding practices, but the implementation of this Initiative was highly questionable due to a lack of routine monitoring and evaluation, which might reduce EIB rates.^{16 17} ((See Page 15 Line 45 -56)

Page 11 line 37 perspective should be changed to prospective

We are sorry for this mistake. Change was made accordingly. (See Page 12 Line 50-51)

Page 11 Line 52 previous studies in China (please provide the reference here)
Thank you for this suggestion. We added Reference 18 to the text. (See Page 13 Line 16-17)

Page 12 line 45 remove “the” developing countries
This grammar mistake was corrected, thank you. (See Page 14 Line 11-12)

Page 12 Line 45 what about other countries that border China eg Vietnam, Laos, Myanmar, Bhutan?
Are the results consistent with these?

We modified this part by adding references from countries bordered China (Laos¹⁰ and Vietnam¹¹) and also from a WHO Global survey conducted mostly in developing countries, including those closed to China (Cambodia, India, Philippines, Thailand, etc.).¹² This part was presented as:

“The positive association between maternal education and early initiation is consistent with studies conducted in the developed countries.^{8,9,13} Previous studies in developing countries showed inconsistent results concerning the relationship between maternal education and early initiation of breastfeeding.^{10-12,14} A study conducted in Laos found that early initiation of breastfeeding was most prevalent among mothers with higher education¹⁰ and a study in Nepal observed the similar result.¹⁴ A study conducted in Vietnam, however, found that mothers with no education were more likely to initiate breastfeeding.¹¹ Results from the WHO Global Survey that was conducted mostly in developing countries found an overall negative association between maternal education and early initiation of breastfeeding.¹²” (See Page 14 Line 9 - 25)

Page 12 line 53-54 remove “with have”
“Have” was removed, sorry for the grammatic mistake. (See Page 14 Line 27-28)

Page 13 line 13 – doctors and nurses fostering initiation of breastfeeding – is there a Baby Friendly Hospital Initiative?

Baby Friendly Hospital Initiative covered most of the hospitals with obstetrical department in China and have had positive impacts on breastfeeding practices, but the implementation of the content in Initiative was still questionable^{16,17}, as there is little monitoring and evaluation to check the quality of the baby friendly hospitals. We briefly discussed Baby Friendly Hospital at the end of Discussion part (please refer to our response to previous comments).

Page 13 Line 60 remove word “trend” as it implies longitudinal analysis
“Trend” was replaced by “association”. (Page 15 Line 34 - 35)

Page 14 line 11- please explain what is meant by ‘routine and successful’ breastfeeding?
This refers to the breastfeeding practices suggested by The American Academy of Pediatrics, which was widely recognized as the optimal practices for breastfeeding.²⁵
We added this citation for this part: “Routine and successful breastfeeding, as per The American Academy of Pediatrics recommendation,²⁵.....” (See Page 16 Line 19 - 25)

Your discussion around traditional perceptions and social norms is good but a stronger discussion of the context that may affect breastfeeding such as policies like the Baby Friendly Hospital Initiative practiced in China? Are the rules and regulations around marketing of BMS etc
Also what impact has the recent melamine contamination crisis had on breastfeeding in China?
Please see the previous detailed response to the similar comment. We added Baby Friendly Hospital Initiative and infant marketing formula issues at the end of discussion part.
We added: “Sociocultural norms in China, such as postpartum confinement, were widely practiced, which might also affect mothers’ breastfeeding behaviors.¹⁵ In addition, despite a nation-wide melamine milk scandal in 2008, the infant formula marketing was even more prevalent in China^{18,19}, after the Chinese government abolished the International Code of Marketing of Breast-milk Substitutes.²⁰ All of these suggest that a large public campaign involving multisectoral stakeholders to

promote breastfeeding practices is highly needed in China.” (Please refer to Page 15 Line 55 -Page 16 Line 7)

Page 14 are there any other opportunities to influence policy or advice on breastfeeding? eg that poorer mums tend to exclusively breastfeeding- how can this be maintained and encouraged? These additions to the discussion will help with implications for planning programs for populations.

We agree that some additional discussion on the policy implications are needed, as suggested by the reviewer. And we discussed more about possible approaches to implement relevant interventions in the last paragraph of the discussion. (See Page 15 Line 40 - Page 16 Line 7).

We also modified the conclusion part to consider the income effects as:

“This study suggests that extensive public health interventions on breastfeeding promotion in China should target on mothers with higher levels of education, especially among those with medium to high incomes. Relevant supportive policies and programs for this group of mothers, including enforcing full-term maternal leave, providing breastfeeding education and counseling, building a mother-friendly workplace, as well as breaking cultural and social norms, should be implemented to create an enabling environment for breastfeeding practice among Chinese mothers. ” (See Page 16 Line 29 - 43)

VERSION 2 – REVIEW

REVIEWER	Marina Cuttini Pediatric Hospital Bambino Gesù, Roma, Italy
REVIEW RETURNED	10-Apr-2019

GENERAL COMMENTS	The Authors have taken into account the reviewer's comments, but there are still some remaining issues. 1. Abstract: possibly in Conclusions the statement "higher education women more likely to early initiate BF" should be completed by something as "but the association was observed only in highest income families". 2. Methods: 2.1 Variable definitions: "EIB prevalence is defined as "proportion of mothers with children...who were put to the breast within one hour...". Why not "proportion of children who...etc etc"? EverBF prevalence: should you add denominator (ie among all children aged 0-12 months)? I feel that a box with definitions that clarify both numerators and denominators would be useful to the Readers. 2.2 Data analyses. 2.2.1 "Data of mothers who did not complete the questionnaire as required were excluded from multivariable analyses": how many were they? Have you considered using multiple imputation techniques, if only as sensitivity analyses (to be provided as supplementary materials)?. 2.2.2 Multivariable analyses: the adjustment according to Zhang and Yu allows to correct the point estimate, but not the CIs (see McNutt et al. Am J Epidemiol 2003,157:940, and The authors reply to Karp, Am J Epidemiol 2014:179(8)1034.) I still believe that it would be better to directly model RRs (see for instance Zou G. Am J Epidemiol 2004; 159 (7): 702). 2.3 Parent involvement. What do you mean by “Investigators do not know the identity of study participants”? By investigators, do you refer to the interviewers only or also researchers? How were the women with a child below 12 months identified? And was informed consent written (with signature by participant) or not? Possibly, answers to this questions should be incorporated in the section “Study design and participants”
---

	2.3 Results. Child's age at interview: as SDs are quite large, it would be of interest to see the full distribution of children age at interview in months (from 0 to 11) by maternal education group (possibly as supplementary table). I feel that measured values for BF variables obtained by the 24h recall method will be dependent on the child's age at interview.
--	--

VERSION 2 – AUTHOR RESPONSE

Response to Reviewer 1.

1. Abstract: possibly in Conclusions the statement "higher education women more likely to early initiate BF" should be completed by something as "but the association was observed only in highest income families".

Thanks for the review to raise this point. This would certainly be a more precise summary of the results.

Thus, we modified this sentence as: "Mothers with a higher education were more likely to initiate early breastfeeding when they were also from a high-income household while also being less likely to exclusively breastfeed their babies." (Please see Page 3 Line 11 - 15)

2. Methods:

2.1 Variable definitions: "EIB prevalence is defined as "proportion of mothers with children...who were put to the breast within one hour...". Why not "proportion of children who...etc etc"?

Ever BF prevalence: should you add denominator (ie among all children aged 0-12 months)?

I feel that a box with definitions that clarify both numerators and denominators would be useful to the Readers.

The definition methods suggested by the reviewer would clarify the outcome variables, Thus, we modified this section per recommendation as:

"EIB prevalence was defined as the proportion of children born in the last 12 months who were put to the breast within an hour after birth. Ever BF prevalence was defined as the proportion of children who were ever breastfed among all children aged 0-12 months." (Please see Page 7 Line 37 - 43).

Also, we added a part at the end of this section to show the numerators and denominators of each variable:

- $EIB = \frac{\text{children aged from 0–12 months who were put to the breast within an hour after birth}}{\text{children aged from 0–12 months}}$
- $EBF = \frac{\text{children aged from 0–6 months who were fed exclusively breast milk in the last 24 hours}}{\text{children aged from 0–6 months}}$
- $PBF = \frac{\text{children aged from 0–6 months who were fed predominantly breast milk in the last 24 hours}}{\text{children aged from 0–6 months}}$
- $\text{Ever BF} = \frac{\text{children aged from 0–12 months who were ever breastfed.}}{\text{children aged from 0–12 months}}$
- $CBF = \frac{\text{children aged from 0–12 months who were breastfed in the last 24 hours.}}{\text{children aged from 0–12 months}}$

(Please see Page 8 Line 5 -18)

2.2 Data analyses.

2.2.1 "Data of mothers who did not complete the questionnaire as required were excluded from multivariable analyses": how many were they? Have you considered using multiple imputation techniques, if only as sensitivity analyses (to be provided as supplementary materials)?.

Thanks for the reviewer to raise this concern. In this study, however, those who did not complete all the questionnaire were classified as non-response population and the researchers did not enter their questionnaire in our database. Thus, we couldn't access the information of those who did not complete the survey. Multiple imputation technique, which is a quite useful method to attenuate the potential bias caused by missing data, could be only applied if we have observed variables.² We, however, did have the data of those who did not complete the questionnaire. Thus, it was therefore not possible to present their characteristics. The total population, including those who did not response to the survey or those who refused to participate after the informed consent process or those who did not complete the questionnaire as requested were less than 10% of the surveyed population. Because the response rate was over 90%, which was a very good response rate for a large scale national representative study, we consider the possibility of potential non-response biases was minimal.

2.2.2 Multivariable analyses: the adjustment according to Zhang and Yu allows to correct the point estimate, but not the CIs (see McNutt et al. Am J Epidemiol 2003,157:940, and The authors reply to Karp, Am J Epidemiol 2014:179(8)1034.)

I still believe that it would be better to directly model RRs (see for instance Zou G. Am J Epidemiol 2004; 159 (7): 702).

Thanks so much for this careful observation, which would definitely improve the robustness and interpretability of the data. After consulting our statistician, we revised all the code and directly modeled RRs, using a generalized linear model with a log link and binomial distribution.³ This model, also called log-binomial or binomial log-linear regression,^{4,5} has been demonstrated by statistical literature to be a good method to estimate RRs when the responses are binomial^{5,6} (all the breastfeeding outcomes in our study only have Yes and No answers).

We revised all the data according to the new model in Table 2. We also corrected the data analysis part as:

“Since the prevalence of the outcomes was high, the odds ratios tended to overestimate the risk ratios. Thus, a log-binomial (or binomial log-linear regression) model^{21 22} was employed to calculate the risk ratios to explore the association between maternal education and breastfeeding outcomes.” Also, we presented some of our RRs, as required by the editor, in the Results section. (Please see Page 9 Line 3 - 12).

2.3 Parent involvement. What do you mean by “Investigators do not know the identity of study participants”? By investigators, do you refer to the interviewers only or also researchers? How were the women with a child below 12 months identified? And was informed consent written (with signature by participant) or not? Possibly, answers to this questions should be incorporated in the section “Study design and participants”

Thanks so much for the reviewer to notice this issue. By investigators, we meant both interviewers and the researchers. Thus, we modified this part as: “Investigators, including both the interviewers and the researchers, do not know the identities of the study participants.” (Please see Page 9 Line 52 -56)

This survey was led by the Chines CDC, where they have access to the birth registration information of all the infants of the region. We added: “Mothers with children under 12 months were identified through birth registration by the China National Center for Disease Control.” in the Study design and participants part. (Please see Page 6 Line 26 -30).

As for the informed consent, we have the relevant information in the Study design and participants part: “. The Institutional Review Board at Peking University Health Science Center and the China National Center for Disease Control approved this study. All participants included in this study provided informed consent..” (Please see Page 6 Line 45-51).

Also, as required by BMJ Open, we put the information at the end of the manuscript as:

“Patient consent

Obtained.

Ethics approval

This study was conducted according to the guidelines laid down in the Declaration of Helsinki and all procedures involving human subjects/patients has been approved by the Institutional Review Boards at Peking University Health Science Center and the China National Center for Disease Control. The participants included in this study provided written informed consent.”

(Please see Page 17 Line 50 – Page 18 Line 5).

2.3 Results.

Child’s age at interview: as SDs are quite large, it would be of interest to see the full distribution of children age at interview in months (from 0 to 11) by maternal education group (possibly as supplementary table). I feel that measured values for BF variables obtained by the 24h recall method will be dependent on the child’s age at interview.

Thanks for the reviewer to raise this point. Demonstrating detailed infant age distribution by maternal education would certainly facilitate the readers to better interpret the findings. Thus, we added Supplement 1 to show the mean, range, median, and percentage distribution of infant age by maternal education.

We also modified the results section as “The mean infant age (in days) for mothers whose highest education was primary school and below, middle school, high/vocational school, and college and above were 186.50 (SD=106.82), 182.11 (SD=104.48), 178.29 (SD=105.82), and 173.10 (SD=105.52), respectively. The detailed distribution of infant age at interview by maternal education can be found in Supplement 1” (Please see Page 10 Line 13 - 23).

Thank you again for kindly considering our paper for potential publication. Should you have any further comments, we are happy to answer them in the due course.

References

1. Yang Z, Lai J, Yu D, et al. Breastfeeding rates in China: a cross-sectional survey and estimate of benefits of improvement. *The Lancet*. 2016;388:S47.
2. Schafer JL. Multiple imputation: a primer. *Statistical methods in medical research*. 1999;8(1):3-15.
3. WACHOLDER S. Binomial regression in GLIM: estimating risk ratios and risk differences. *American journal of epidemiology*. 1986;123(1):174-184.
4. Blizzard L, Hosmer W. Parameter estimation and goodness-of-fit in log binomial regression. *Biometrical Journal*. 2006;48(1):5-22.
5. Greenland S. Model-based estimation of relative risks and other epidemiologic measures in studies of common outcomes and in case-control studies. *American journal of epidemiology*. 2004;160(4):301-305.
6. Barros AJ, Hirakata VN. Alternatives for logistic regression in cross-sectional studies: an empirical comparison of models that directly estimate the prevalence ratio. *BMC medical research methodology*. 2003;3(1):21.